# A Novel Ensemble Learning-Based Computational Method to Predict Protein-Protein Interactions from Protein Primary Sequences

**DOI:** 10.3390/biology11050775

**Published:** 2022-05-19

**Authors:** Jie Pan, Shiwei Wang, Changqing Yu, Liping Li, Zhuhong You, Yanmei Sun

**Affiliations:** 1Key Laboratory of Resources Biology and Biotechnology in Western China, Ministry of Education, College of Life Science, Northwest University, Xi’an 710069, China; jiepan960930@gmail.com (J.P.); wangsw@nwu.edu.cn (S.W.); 2School of Information Engineering, Xijing University, Xi’an 710123, China; 20160082@xijing.edu.cn; 3College of Grassland and Environment Sciences, Xinjiang Agricultural University, Urumqi 830052, China; 4School of Computer Science, Northwestern Polytechnical University, Xi’an 710129, China; zhuhongyou@gmail.com

**Keywords:** protein–protein interaction, Discrete Hilbert transform, rotation forest, position-specific scoring matrices

## Abstract

**Simple Summary:**

Protein–protein interactions (PPIs) play a central role in the evolution and progression of various biological processes. In this article, we constructed a novel ensemble-learning-based model to predict potential PPIs, which only utilized the protein sequence information. The presented method used Discrete Hilbert transform to extract amino acid sequence information from position-specific scoring matrices. Then these extracted features were fed into rotation forest for training and predicting. When applying our method to the three datasets (*Yeast*, *Human*, and *Oryza sativa*) for detecting PPIs, we obtained excellent prediction performance. Furthermore, the comparison results indicated that our computational model is effective and robust in predicting potential PPI pairs.

**Abstract:**

Protein–protein interactions (PPIs) are crucial for understanding the cellular processes, including signal cascade, DNA transcription, metabolic cycles, and repair. In the past decade, a multitude of high-throughput methods have been introduced to detect PPIs. However, these techniques are time-consuming, laborious, and always suffer from high false negative rates. Therefore, there is a great need of new computational methods as a supplemental tool for PPIs prediction. In this article, we present a novel sequence-based model to predict PPIs that combines Discrete Hilbert transform (DHT) and Rotation Forest (RoF). This method contains three stages: firstly, the Position-Specific Scoring Matrices (PSSM) was adopted to transform the amino acid sequence into a PSSM matrix, which can contain rich information about protein evolution. Then, the 400-dimensional DHT descriptor was constructed for each protein pair. Finally, these feature descriptors were fed to the RoF classifier for identifying the potential PPI class. When exploring the proposed model on the *Yeast*, *Human*, and *Oryza sativa* PPIs datasets, it yielded excellent prediction accuracies of 91.93, 96.35, and 94.24%, respectively. In addition, we also conducted numerous experiments on cross-species PPIs datasets, and the predictive capacity of our method is also very excellent. To further access the prediction ability of the proposed approach, we present the comparison of RoF with four powerful classifiers, including Support Vector Machine (SVM), Random Forest (RF), K-nearest Neighbor (KNN), and AdaBoost. We also compared it with some existing superiority works. These comprehensive experimental results further confirm the excellent and feasibility of the proposed approach. In future work, we hope it can be a supplemental tool for the proteomics analysis.

## 1. Introduction

Predicting protein–protein interactions (PPIs) is essential for elucidating protein functions and understanding the biological structures in cells [1]. Additionally, the prediction of PPIs not only helps people to further examine how proteins exert their various functions, but also provides the crucial information for the design of targeted drugs. in the past decade, there have been many biological experimental approaches, including mass spectrometry [2], tandem affinity purification [3], and two-yeast hybrids [4] have been extensively studied for decades. However, these conventional studies present some drawbacks, such as high cost, time-intensive, and suffer from high rate of false-positives and false-negatives. Accordingly, the development of novel computational approaches to predict potential PPI pairs would be of enormous value to biologists [5].

To date, several computational methods for PPIs’ prediction have been presented. In general, these methods can be broadly grouped into three types: ligand-based approaches, structure-based methods, and sequence-based methods. Typically, the sequence-based methods do not perform as well as the first two methods, while the ligand and structure-based approaches usually need the a priori information of proteins. The challenging problem will arise when this information did not exist. In recent years, following the advancement of genome technologies, a large amount of protein sequence data had been collected and entered in databases. Therefore, the sequence-based methods to identify PPIs have aroused an increasing concern. The vast majority of the existing computational methods are usually based on the machine learning algorithms, including rotation forest [6], support vector machine [7,8], and Naive Bayes [9]. For example, Huang et al. [10] adopted discrete cosine transform descriptors and weighted sparse representation model to predict PPIs from protein sequence. You et al. [11] proposed a method called PCA-EELM, which utilized four different types of sequence information to predict PPIs. Li et al. [12] proposed a method called PSIPEL that combined an novel feature extraction approach, Low Rank Approximation with Rotation Forest, to predict PPIs from protein primary sequences. Zeng et al. [13] developed a deep learning framework to predict PPIs, which employed a sliding window and text convolutional neural network to capture local contextual and global sequence features from target proteins, respectively. Chen et al. [14] applied Fast Fourier Transform to capture protein feature descriptors and fed them to Random Projection for training and detecting self-interacting proteins [15]. Different from the traditional machine learning-based methods, deep learning-based approaches can not only extract feature vectors from the protein sequence directly, but also can capture their nonlinear relationships to improve the prediction performance. As a consequence, deep learning algorithms also have been widely employed in PPI prediction in recent years. For example, Sun et al. [16] first adopted a deep learning technique, stacked autoencoder, for predicting human PPIs from amino acid sequence. Zhang et al. [17] presented Ensemble Deep Neural Networks (EnsDNN), which is a neural network-based method that employs different protein descriptors to detect PPIs. Yao et al. [18] designed a novel method called Res2vec to represent protein sequences, then the residual representation was integrated into a deep neural network for training and predicting. Hashemifar et al. [19] developed a method named DPPI, which combined data augmentation, convolutional neural network, and random projection to predict PPIs. Richoux et al. [20] made a comparison of two powerful deep learning models and discussed the required attention when applying the deep learning algorithm to PPI prediction. Despite of these achievements, there is still great room for these computational based approaches to attain improvement [21].

Inspired by these excellent works, we herein attempted to develop a new computational model to predict potential PPIs from the information of amino acid sequences. Specifically, we first transformed the sequences into a position-specific scoring matrix (PSSM), from which we could preserve the evolution information of primary protein sequence. Then the Discrete Hilbert transform (DHT) algorithm was adopted to capture feature descriptors from the PSSM. Finally, the Rotation Forest (RoF) classifier was used for training and determining whether the proteins are related or not. In order to access the predictive ability of our approach, we performed it on the *Yeast*, *Human,* and *Oryza sativa* PPIs datasets, and yielded a high prediction accuracy of 91.93, 96.35, and 94.24%, respectively. Moreover, we compared our approach with several existing sequence-based methods. We also applied it on four independent PPI datasets. Experimental results demonstrated that our method is effective for identifying whether the protein pairs interact or not, and it can be considered as a supplemental tool to the commonly used experimental methods.

## 2. Materials and Methodology

### 2.1. Protein Interaction Dataset

In this article, the presented approach was first validated on a high-confidence PPIs dataset named *Yeast*, which was selected from the Database of Interaction Proteins (DIP) [22] by Guo et al. [23]. This dataset was collected from the *Saccharomyces cerevisiae* core subset which contains 5996 interaction pairs. In order to remove redundant information, the CD-Hit [24,25] was employed in this work. CD-Hit is a multiple sequence alignment tool for removing the homologous sequence pairs. After removing the protein pairs which had ≥40% sequence identity or the fragments with less than 50 residues, we obtained 5594 protein pairs as the positive samples. For the construction of a negative dataset, we randomly chose 5594 additional *Yeast* pairs from different subcellular compartments. Accordingly, the final *Yeast* PPIs dataset contained 11,188 protein pairs.

To indicate the generality of the proposed approach, we also verified our experiment on the *Human* and Rice (*Oryza sativa*) PPIs dataset. The *Human* dataset was selected from the Human Protein Reference Database (HPRD) [26]. After removing sequences with greater than 25% sequence identity, we employed 3899 interaction pairs, which collected from 2502 different human proteins to construct the positive samples. For the negative samples, we used the same approach to construct the negative samples of Human dataset. Finally, the negative set consisted of 4262 pairs from 661 proteins. In addition, *Oryza sativa* dataset was collected from the PRIN [27] database. The *Oryza sativa* dataset is consists of 4800 positive samples and 4800 negative samples.

### 2.2. Encoding Amino Acid Sequence as Date Matrix

The Position-Specific Scoring Matrix (PSSM) was adopted to represent the protein sequence. It was presented by Gribskov et al. [28] to analysis the sequence similarities of proteins. PSSM produces excellent results in many fields, such as in protein secondary structure prediction [29], disorder region prediction [30], and DNA function prediction [31]. A PSSM is a matrix that can be represented as $PSSM=\left\{\varphi_{m,n}:m=1\cdots b\;\mathrm{and}\;n=1\cdots 20\right\}$, where *m* denotes the length of the protein sequence, and the number 20 represents the 20 amino acids. The $\varphi_{m,n}$ can be expressed as follows:(1)$$\varphi_{m,n}=\sum_{t=1}^{20}P(a,q)\times w(b,q),\;a=1\cdots P,\;b=1\cdots 20$$
where P(a,q) indicate the frequency value of the qth amino acid at the position a of the probe, and w(b,q) indicate the value of Dayoff mutation matrix between the acid of bth and qth. The main concern in applying the PSSM algorithm is that it can enable the sequence to match the alignment table by awarding a higher score to a conservative position, while a good score means a conservative position and a low score represents a low-conserved position.

In this work, the PSI-BLAST tool was applied to transform the protein sequence into a PSSM matrix. BLAST is a useful resource for searching local similarity regions between different amino acid sequences. It can make a comparison of sequences and nucleotides with particular databases, and compute a statistical significance of the matches, to infer functions and evolutionary associations between different sequence. PSI-BLAST is an enhanced BLAST technique, which can robustly identify novel proteins in distantly related organisms. The main improvement of PSI-BLAST is that it can adopt the profile to search the non-redundant *SWISS-PROT* database, and then employ the searched results to rebuild the profile, and so on, until no new results are generated. *SWISS-PROT* is an annotated protein sequence database and the sequences collected in it are searched for by many authoritative biologists. Moreover, to better exploit the performance of the PSI-BLAST algorithm, we chose three iterations, and the e-value parameter was assigned to 0.001, and the PAM was selected as the scoring matrix. The other parameters were set to their default values.

### 2.3. Discrete Hilbert Transform

In this work, the Discrete Hilbert transform [32] (DHT) algorithm was adopted to capture feature values from the PSSM matrix to generate the feature vectors, which can make the prediction results more accurate. Discrete Hilbert transform was first employed to analysis the signal in the frequency and time domains. Before introducing the 2-D DHT, the 1-D DHT is first used in spatial and frequency domain. Let ℓ(a) represent the discrete signal, $\overset{\wedge }{\ell }(a)$ can be shown as:(2)$$\overset{\wedge }{\ell }(a)=\ell (a)*p(a)$$
where:(3)$$p(a)=\frac{1-{\left(-1\right)}^{a}}{a\pi }\;a=1,2,3\cdots$$

After applying the Fourier transform (FT), $\overset{\wedge }{\ell }(a)$ could be represented as:(4)$$\overset{\wedge }{\ell }(a)=\mathrm{IDFT}\left[\overset{\wedge }{F}(j\Omega )\right]=\mathrm{IDFT}\left[F(j\Omega )-\left[-j\mathrm{sgn}(\Omega )\right]\right]$$

In Equation (4), *IDFT* represents the Inverse Discrete Fourier transform [33], and the Fourier transform of ℓ(a) and $\overset{\wedge }{\ell }(a)$ can be described as $\overset{\wedge }{F}(j\Omega )$ and F(jΩ), respectively. Above all, the function H(jΩ) can be written as:(5)$$H(j\Omega )=-j\mathrm{sgn}(\Omega )=\left\{\begin{array}{ll} -j & \Omega >0,\\ j & \Omega <0, \end{array}\right.$$
where angular frequency is Ω and the finite discrete signum function is denoted by sgn(Ω). For better capturing feature vectors from the PSSM matrix, we applied the 2D DHT [34] that was defined in the frequency domain to extract features from the PSSM. The odd and even parts of the PSSM features in the frequency domain refer to the highly conserved order of amino acids within a particular protein sequence. Suppose that the odd and even parts of PSSM features in a frequency domain are defined by $f_{0}(x,y)$ and $f_{e}(x,y)$, respectively. The formula of the 2D Discrete Hilbert transform can be written as:(6)$$f_{0}(x,y)=\left[\mathrm{sgn}(x,y)+bdy(x,y)\right]f_{e}(x,y)$$
(7)$$\mathrm{sgn}(x,y)=\left\{\begin{array}{llll} 1 & 0<x<\frac{1}{2}, & 0<y<\frac{H_{2}}{2}\\ -1 & \frac{H_{1}}{2}<x<H_{1}, & \frac{H_{2}}{2}<y<H_{2}\\ 0 & \mathrm{elsewhere} & & \end{array}\right.$$
where bdy(x,y) is employed to adjust the boundary and the finite discrete signum function is described by sgn(x,y). $H_{1}$ and $H_{2}$ represent the size of $f_{0}(x,y)$ and $f_{e}(x,y)$, respectively. Given an image P(x,y), the 2D DHT of $\overset{\wedge }{P}(x,y)$ in the frequency domain can be expressed as:(8)$$\overset{\wedge }{P}(x,y)=\left[\mathrm{sgn}(x,y)+bdy(x,y)\right]\cdot P(x,y)$$
where $x=0,1,\cdots ,H_{1}-1$ and $y=0,1,\cdots ,H_{2}-1$; $H_{1}$ and $H_{2}$ are the size of the input image.

Let T(i,j) represent an image, then the 2D DHT of T(i,j) in spatial domain can be defined as:(9)$$\overset{\wedge }{T}(i,j)=T(i,j)*R(i,j)$$
(10)$$R(i,j)=\left(\cot\left(\frac{\pi }{H_{1}}\right)i+\cot\left(\frac{\pi }{H_{2}}\right)j\right)\frac{2}{H_{1}H_{2}}$$
where $i=0,1,2,\cdots ,H_{1}\cdots 1$ and $j=0,1,2,\cdots ,H_{2}-1$. Because of the same mathematic principle between 1D and 2D DHT, the image f(i,j) can be expanded as a 2D Fourier series:(11)$$f(i,j)=\frac{1}{ZQ}\sum_{u=0}^{Z-1}\sum_{v=0}^{Q-1}\left|F(\alpha ,\beta )\right|\sin(\phi_{\alpha ,\beta }(i,j))$$
where $F(\alpha ,\beta )=\sum_{i=0}^{Z-1}\sum_{j=0}^{Q-1}f(i,j)e^{-j2\pi (\frac{\alpha i}{Z}+\frac{\beta j}{Q})}$, α=0,1,2,⋯,Z−1, and i=0,1,2,⋯,Q−1; *Z* and *Q* are the size of the image.

### 2.4. Ensemble-Learning-Based Classifier

Rotation forest (RoF) is an ensemble learning algorithm, which was introduced by Rodriguez et al. [35] to improve the diversity and accuracy of each classifier in the ensemble system. The main contribution of the RoF algorithm is that it applies the principal component analysis (PCA) technique to construct a rotational matrix, which can then transform initial variables into new variables to construct new independent decision trees. Moreover, PCA algorithm ensures the diversity of the classifier, and it retains most of the evolutionary information of the protein feature descriptors [36]. The specific framework of this algorithm is summarized as follows.

Let *T* represents the training sample set, *H* denotes the feature set, and *E* be the corresponding labels. Let α be the set of class labels $\left\{\alpha_{1},\alpha_{2}\right\}$, from which *E* takes values. Assume that *T* is a N×n matrix, where *n* and *N* represents the features and training samples in the PPIs data set. The data will be divided randomly into *K* subsets of the approximate size; there are *L* decision trees represented as $D_{1},\ldots ,D_{L}$, respectively. In the RoF algorithm, *L* and *K* are the two parameters that require advance optimization. The specific details of the RoF algorithm can be defined as follows:

(1) Divide the feature set *H* optionally into *K* subsets. Assume that *K* is a factor of *m*, then, each feature will include u=m/K features.

(2) Let $H_{ij}$ represent the *j*-th subset of features for training classifier $D_{i}$. The features of dataset *T* in $H_{ij}$ is defined as $T_{ij}$. Then a bootstrap subset of size 75% of the data set is extracted to construct the training set, which is defined as $T_{ij}{}^{\prime }$. Then the PCA algorithm is adopted with $T_{ij}{}^{\prime }$ to generate the coefficients into a matrix $C_{ij}$. Denoted as $a_{ij}^{\left(1\right)},\cdots ,a_{ij}^{\left(M_{i}\right)}$, the size of each $T_{ij}{}^{\prime }$ is U×1.

(3) Using the coefficients in $C_{ij}$ to build a spare rotation matrix $R_{i}$ and it can be expressed as follows;
(12)$$R_{i}=\left[\begin{array}{cccc} a_{i1}^{\left(1\right)},\cdots ,a_{i1}^{\left(M_{1}\right)} & \left\{0\right\} & \cdots & \left\{0\right\}\\ \left\{0\right\} & a_{i1}^{\left(1\right)},\cdots ,a_{i2}^{\left(M_{2}\right)} & \cdots & \left\{0\right\}\\ \left\{0\right\} & \left\{0\right\} & \cdots & a_{iK}^{\left(1\right)},\cdots ,a_{iK}^{\left(M_{K}\right)} \end{array}\right]$$

In the classification stages, provided there is a target sample *x*, let $d_{ij}(XR_{i}^{a})$ denotes the probability produced by the classifier $D_{i}$ to the class $\alpha_{i}$. Finally, the confidence level of each class can be found through the mean combination technique:(13)$$\lambda_{i}(x)=\frac{1}{L}\sum {}_{i=1}^{L}d_{ij}(xR_{i}^{a})$$

In this way, the test sample *x* can 190 be easily distributed to the class with the highest confidence.

## 3. Results

### 3.1. Evaluation Measures

In this study, in order to avoid over-fitting to affect the predictive ability of the proposed method, we used the five-fold cross-validation (five-fold CV) technique to measure the predictive ability of the proposed method. All samples were randomly split into five subsets, in which four were used as a training set and the other one was adopted as the test set. In this experiment, this procedure was performed five times to guarantee that each subset was used once as a test subset. Lastly, the average and standard deviations of these five experiments were taken as the final experiment results. In our experiments, several evaluation criteria were employed to estimate the predictive ability of the proposed model, including accuracy (*ACC*), sensitivity (*Sen.*), specificity (*Spec.*.), precision (*PR*), and Matthews’ correlation coefficient *(MCC)* to access the predictive power. Their corresponding calculating formulae are as follows:(14)$$ACC=\frac{TP+TN}{TN+FP+TP+FN}$$
(15)$$Sen.=\frac{TP}{FN+TP}$$
(16)$$Spec.=\frac{TN}{TN+FP}$$
(17)$$PR=\frac{TP}{FP+TP}$$
(18)$$MCC=\frac{TN\times TP-FN\times FP}{\sqrt{\left(TP+FN)\times (TP+FP)\times (ΤΝ+FP)\times (TN\times FN\right)}}$$
where true positive (*TP*) indicated the quantity of true samples, which can be identified correctly; false positive (*FP*) represents the amount of true non-interacting pairs detected to be PPIs falsely; true negative (*TN*) is the amount of true non-interacting pairs that are correctly identified; false negative (*FN*) represents the number of true samples categorized as non-interacting pairs incorrectly. Additionally, the receiver operating characteristic (ROC) curves were also plotted in order to prove the predictive power of our method. The AUC (area under ROC curves) values were also calculated to express the ROC values in a more accessible way.

### 3.2. Prediction Performance on Three PPIs Datasets

In this study, we first validated our model on the *Yeast* data set, and Table 1 summarizes the results of the five-fold cross-validation (five-fold CV) experiment. It can be seen from Table 1 that the average accuracy, sensitivity, specificity, precision, and MCC values are 91.93%, 89.78%, 94.05%, 93.82%, and 85,14%, and their standard deviations were 0.69%, 0.79%, 1.30%, 1.19%, and 1.15%, respectively. Then, the proposed method was performed on the *Human* PPIs dataset; we also yielded excellent predicted results shown on Table 2, with average accuracy, sensitivity, specificity, precision, and MCC values of 96.35%, 95.76%, 96.87%, 96.57%, and 92.95%, and their standard deviations were 0.56%, 0.78%, 0.71%, 0.64%, 1.03%, respectively. In addition, to further demonstrate the robustness of the proposed model, we finally applied it to a plant PPI dataset, *Oryza sativa*. With respect to the *Oryza sativa* dataset, the average accuracy, sensitivity, specificity, precision, and MCC values of the proposed model are shown in Table 3 as 94.24%, 94.50%, 94.02%, 94.02%, and 89.14%, and their standard deviations were 0.37%, 0.97%, 0.82%, 1.03%, and 0.66%, respectively. The receiver operating characteristic (ROC) curves for the three benchmark datasets are shown in Figure 1, Figure 2 and Figure 3. We also calculated the area under the ROC curve (AUC) values of these three PPI datasets for further evaluate the predictive power of our model, and they were 0.9586, 0.9831, and 0.9667, respectively.

### 3.3. Compared with Different Classifier Models

To date, there are a lot of machine learning algorithms have been developed for detecting PPIs. To further verify the prediction accuracy of the proposed model, we compared it with some popular classifiers, including Support vector machine (SVM), Random Forest (RF), K-Nearest Neighbor (KNN), and AdaBoost algorithm. To be specific, we utilized the same DHT descriptors and compared the predictive performance between RoF and these classifiers. We used the LIBSVM tool to train and predict the SVM-based model. To optimize the best parameter of the SVM classifier, the grid search method was adopted to select the best parameters of SVM c and g. We set c = 13, g = 0.0006 and c = 3, g = 0.0005 for the *Yeast* and *Human* data set. When performing on the *Oryza sativa* data set, we set c = 7, g = 0.0009. The parameter K of RF classifiers of the *Yeast*, *Human*, and *Oryza sativa* dataset were 27, 7, and 17, respectively. The parameters of KNN model included the number of neighbors and distance measures. In this article, all the experiments used the Manhattan distance, and the number of neighbors for these three PPI data sets were 15, 17, and 4, respectively. Table 4 illustrated the details of the prediction results of these four state-of-art classifiers on the *Yeast*, *Human*, and *Oryza sativa* data set. To identify any potential overfitting or underfitting problems in the proposed model, we also used a train/test/validation process for predicting these datasets. The experimental results based on this approach can be seen in our Appendix A.

As shown in Table 4, the proposed method provided the best results on the three PPI data sets in terms of all the metrics, and the least accuracy improvement was reached with 7.49% on the *Yeast* dataset, 1.03% on the *Human* data set, and 8.66% on the *Oryza sativa* data set. The lowest enhanced AUC values were reached with 3.34% on the *Yeast* dataset, 0.13% on the *Human* dataset, and 4.17% on the *Oryza sativa* dataset. For the visual analysis, we drew a histogram for the ACC and AUC values that were generated by these powerful classifiers in Figure 4. These experimental results further demonstrated that rotation forest is the best classifier for the features that we introduced.

### 3.4. Evaluation of Prediction Ability on Four Independent Dataset

Although the proposed model has achieved satisfactory results on the *Yeast*, *Human,* and *Oryza sativa* PPI datasets, we also applied it on four independent datasets, including *H. sapiens*, *H. pylori*, *M. muscules,* and *C. elegans,* to further demonstrate the suitability of our method. In this experiment, we utilized all of the *Yeast* dataset as the training set and the other four independent datasets were used as the test sets in order to verify the robustness of the proposed method. In addition, we also compared the predictive performance with some excellent approaches. Table 5 summarizes the results of the accuracy comparisons between our model and some existing methods on the four datasets. It can be seen that the prediction accuracy yielded by our method on the *H. sapiens*, *H. pylori*, *M. muscules,* and *C. elegans* datasets were all higher than 91%, which were 94.27, 91.67, 93.12, and 92.14%, respectively. These experimental results further indicated that our method has strong a generalization ability to predict PPIs. (N/A means not available.)

### 3.5. Compared with Existing Methods

In recent years, various kinds of computational methods have been proposed for predicting potential protein–protein interactions. Here, we compared the prediction ability of the proposed model with some popular methods on the *Yeast* and *Human* dataset, which were also utilized in the five-fold cross-validation method. Table 6 and Table 7 list the predictive performance of these methods with several common evaluation criteria, including accuracy, precision, sensitivity, and MCC. From Table 6, we can see that our method produced an accuracy of 91.93% on the *Yeast* dataset; the precision is 93.82%, the sensitivity is 89.78%, and the MCC value is 85.14%. The average accuracy results of selected methods are all lower than our method on the *Yeast* dataset. Table 7 summarizes the average results of these collected approaches on the *Human* dataset, and are between 90.57 and 96.09%, while the average accuracy of our method is as high as 96.35%. These results further indicated that combining the DHT descriptor and rotation forest classifier is effective for PPIs’ prediction. (N/A means not available.)

## 4. Discussion

The identification of protein–protein interactions (PPIs) can provide a novel perspective for clinical diagnosis and treatment. It also plays an important role in inter-cellular and intra-cellular functions and inter-molecular connectivity. In this article, we presented a novel ensemble-learning-based method to predict potential PPIs that only used the amino acid sequence information. There are four reasons why the proposed model has excellent prediction performance. First, all protein sequence data were preprocessed to remove residues and redundant information. Second, the target protein sequences were calculated into features by the PSSM technique, which can embed the evolutionary information in the form of a matrix. Thirdly, the Discrete Hilbert transform (DHT) algorithm was employed to extract the feature descriptors from the PSSM. In this way, the proposed model can capture high-dimensional and complex potential information to improve the prediction performance. Finally, the ensemble-learning-based classifier, rotation forest (RoF), was utilized to deal with the classification problem. We performed our method on three PPIs datasets (*Yeast*, *Human* and *Oryza sativa*) under five-fold cross-validation. To further demonstrate the excellent prediction ability of our method, we also applied it in four independent cross-species datasets and compared it with some existing excellent methods. The comprehensive experimental results indicated that our model can be served as a powerful tool to guide researchers to study the functions and roles of proteins. However, there are still some limitations in our work. Firstly, the negative datasets were the random section from the non-interacting pairs. These negative sets may include false negative cases. This has the potential to affect the prediction accuracy of the developed model. In future work, we will investigate the DHT algorithm, which is more appropriate for problems involving large feature dimensions and a small number of training samples; through this, we are hoping to better solve the problem of protein–protein interaction prediction.

## 5. Conclusions

In this study, we proposed a novel ensemble learning based model that can greatly improve sequence-based PPIs’ prediction. We conducted a comprehensive experiment on three gold standard datasets. Furthermore, we performed independent validation on four cross-species PPI datasets. Experimental results based on cross validations and comparison indicated that our method is effective and robust in predicting PPIs.

## Figures and Tables

**Figure 1 biology-11-00775-f001:**
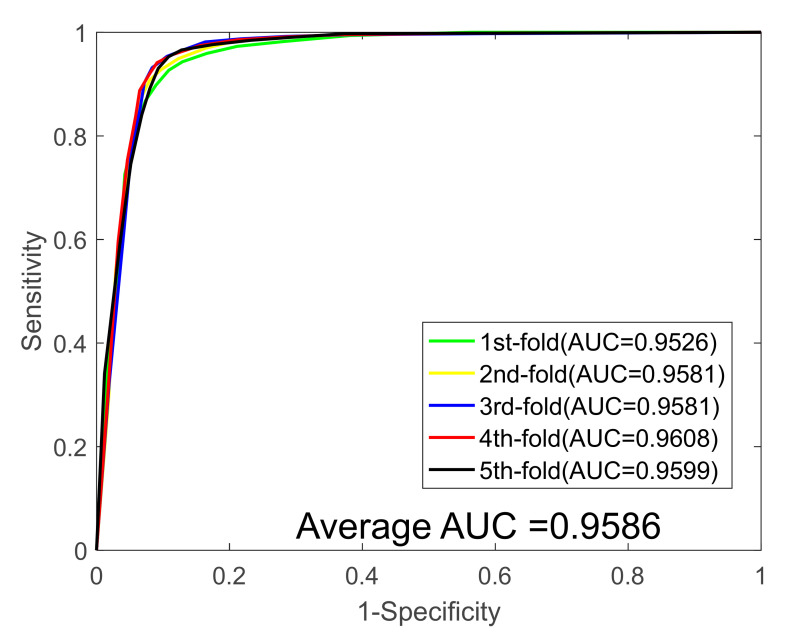
ROC curves generated by the proposed model on the *Y**east* dataset.

**Figure 2 biology-11-00775-f002:**
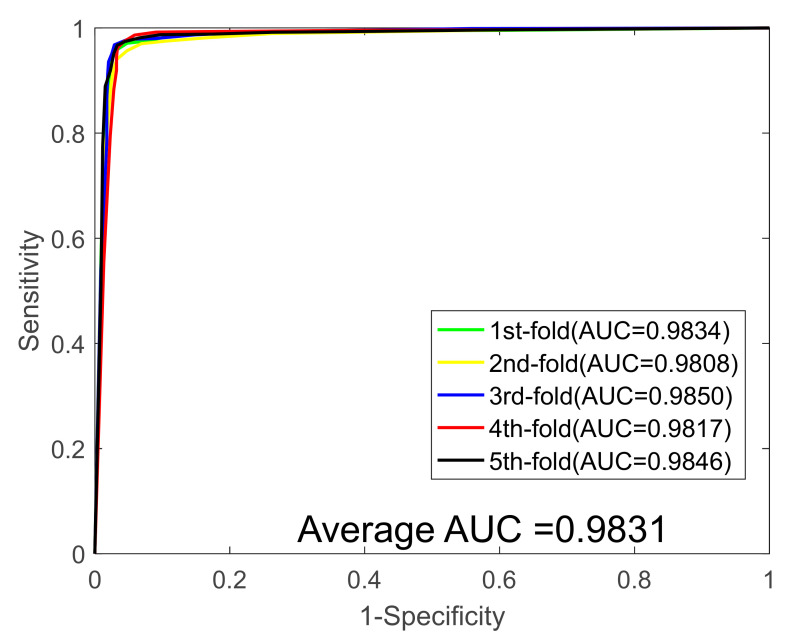
ROC curves generated by the proposed model on the *Human* dataset.

**Figure 3 biology-11-00775-f003:**
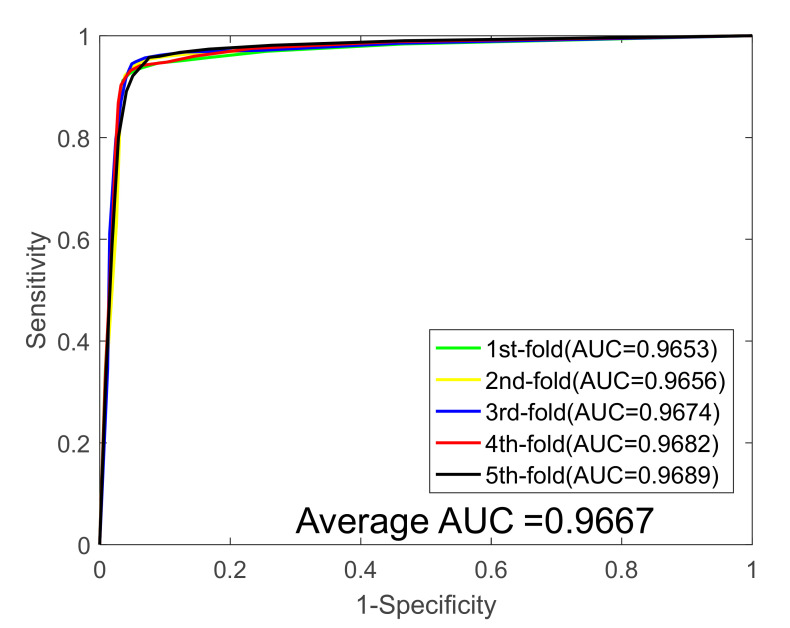
ROC curves generated by the proposed model on the *Oryza sativa* dataset.

**Figure 4 biology-11-00775-f004:**
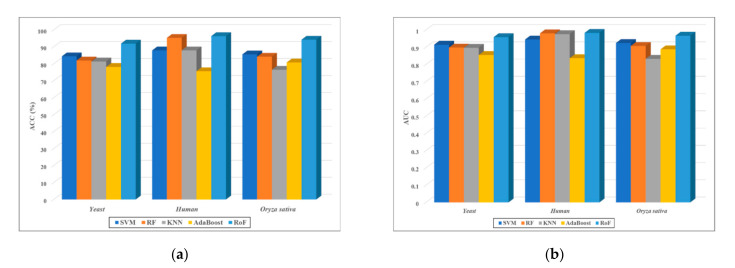
Comparison of the results produced by different classifier models on three benchmark datasets. (**a**) Is the obtained accuracy results on the three benchmark datasets. (**b**) Is the obtained AUC results on the three benchmark datasets.

**Table 1 biology-11-00775-t001:** Five-fold CV results performed by the proposed model on the *Yeast* PPIs dataset.

Dataset	ACC (%)	Sen. (%)	Spec. (%)	PR (%)	MCC (%)	AUC
1	90.88	89.13	92.67	92.57	83.42	0.9562
2	91.55	90.31	92.79	92.55	84.52	0.9581
3	92.40	89.42	95.37	95.04	85.93	0.9581
4	92.49	90.90	94.15	94.20	86.10	0.9608
5	92.31	89.15	95.30	94.73	85.75	0.9599
Average	91.93 ± 0.69	89.78 ± 0.79	94.05 ± 1.30	93.82 ± 1.19	85.14 ± 1.15	0.9586 ± 0.0018

**Table 2 biology-11-00775-t002:** Five-fold CV results performed by the proposed model on the *Human* PPIs dataset.

Dataset	ACC (%)	Sen. (%)	Spec. (%)	PR (%)	MCC (%)	AUC
1	96.20	95.23	97.08	96.72	92.67	0.9834
2	95.47	95.23	95.69	95.47	91.34	0.9808
3	96.94	97.10	96.78	96.62	94.06	0.9850
4	96.63	95.73	97.40	96.89	93.44	0.9817
5	96.51	95.53	97.41	97.14	93.24	0.9846
Average	96.35 ± 0.56	95.76 ± 0.78	96.87 ± 0.71	96.57 ± 0.64	92.95 ± 1.03	0.9831 ± 0.0018

**Table 3 biology-11-00775-t003:** Five-fold CV results performed by the proposed model on the *Oryza sativa* PPIs dataset.

Dataset	ACC (%)	Sen. (%)	Spec. (%)	PR (%)	MCC (%)	AUC
1	93.91	94.64	93.22	92.94	88.55	0.9635
2	94.38	94.64	94.13	93.83	89.38	0.9656
3	94.79	95.09	94.48	94.70	90.12	0.9674
4	94.22	95.28	93.17	93.22	89.10	0.9628
5	93.91	92.84	95.08	95.40	88.54	0.9689
Average	94.24 ± 0.37	94.50 ± 0.97	94.02 ± 0.82	94.02 ± 1.03	89.14 ± 0.66	0.9667 ± 0.0022

**Table 4 biology-11-00775-t004:** Predictive performance comparison among four different classifiers.

Dataset	Method	ACC (%)	Sens. (%)	Spec. (%)	PR (%)	MCC (%)	AUC
*Yeast*	SVM	84.44 ± 0.84	83.14 ± 1.01	85.77 ± 1.40	85.37 ± 1.68	73.71 ± 1.17	0.9149 ± 0.0061
	RF	81.97 ± 0.41	80.26 ± 1.27	83.68 ± 0.48	83.09 ± 0.84	70.41 ± 0.55	0.8979 ± 0.0038
	KNN	81.39 ± 1.07	75.19 ± 2.16	87.63 ± 1.17	85.88 ± 1.21	69.47 ± 1.37	0.8967 ± 0.0057
	AdaBoost	78.15 ± 1.82	76.88 ± 1.90	79.45 ± 2.95	85.46 ± 2.85	65.87 ± 1.97	0.8546 ± 0.0120
	RoF	91.93 ± 0.69	89.78 ± 0.79	94.05 ± 1.30	93.82 ± 1.19	85.14 ± 1.15	0.9586 ± 0.0018
*Human*	SVM	87.93 ± 0.86	85.78 ± 1.28	89.89 ± 1.37	88.59 ± 1.53	78.69 ± 1.31	0.9446 ± 0.0069
	RF	95.32 ± 0.96	92.63 ± 1.93	97.82 ± 0.94	97.50 ± 1.03	91.04 ± 1.74	0.9804 ± 0.0016
	KNN	87.92 ± 1.19	76.67 ± 2.44	98.23 ± 0.49	97.51 ± 0.74	78.10 ± 1.96	0.9758 ± 0.0046
	AdaBoost	75.64 ± 1.69	71.36 ± 3.87	79.53 ± 3.04	76.19 ± 2.29	62.88 ± 1.83	0.8362 ± 0.0170
	RoF	96.35 ± 0.56	95.76 ± 0.78	96.87 ± 0.71	96.57 ± 0.64	92.95 ± 1.03	0.9831 ± 0.0018
*Oryza* *sativa*	SVM	85.58 ± 1.27	84.06 ± 1.08	87.16 ± 2.46	86.73 ± 2.63	75.32 ± 1.78	0.9246 ± 0.0085
	RF	84.19 ± 0.92	81.71 ± 1.23	86.68 ± 1.03	85.99 ± 0.85	73.34 ± 1.24	0.9070 ± 0.0096
	KNN	76.51 ± 0.70	85.19 ± 0.88	67.82 ± 0.91	72.58 ± 1.10	63.50 ± 0.77	0.8327 ± 0.0040
	AdaBoost	80.82 ± 1.37	81.50 ± 1.87	80.16 ± 1.61	80.40 ± 2.00	69.01 ± 1.67	0.8876 ± 0.0132
	RoF	94.24 ± 0.37	94.50 ± 0.97	94.02 ± 0.82	94.02 ± 1.03	89.14 ± 0.66	0.9667 ± 0.0022

**Table 5 biology-11-00775-t005:** Prediction accuracy of the four independent datasets.

Species	Test Pair	Our Method	Ding et al. [37]	Huang et al. [10]	Zhan et al. [38]	Wang et al. [39]
*H. sapiens*	1412	94.29%	90.23%	82.22%	91.93%	80.10%
*H. pylori*	1420	91.67%	90.34%	82.18%	91.34%	N/A
*M. muscules*	313	93.12%	91.37%	79.87%	94.89%	89.14%
*C. elegans*	4013	92.14%	86.72%	81.19%	93.20%	92.96%

**Table 6 biology-11-00775-t006:** Performance comparisons of computational methods on the *Yeast* dataset.

Author	Method	ACC (%)	PR (%)	Sens. (%)	MCC (%)
Guo et al. [23]	ACC + SVM	89.33	89.93	88.87	N/A
Yang et al. [40]	LD + KNN	86.15	90.24	81.30	N/A
Wang et al. [41]	3-MER + CNN	90.26	91.65	88.14	82.38
Zhou et al. [42]	LD + SVM	88.56	89.50	87.37	77.15
An et al. [43]	PSSMMF + SVM	90.48	90.58	90.26	82.84
You et al. [11]	PCA + ELLM	87.00	87.59	86.15	77.36
Our method	DHT + RoF	91.93	93.82	89.78	85.14

**Table 7 biology-11-00775-t007:** Performance comparisons of computational methods on the *Human* dataset.

Author	Method	ACC (%)	PR (%)	Sens. (%)	MCC (%)
Ding et al. [37]	MMI + RF	96.08	96.67	95.05	92.17
Li et al. [44]	OLPP + RoF	96.09	96.56	95.20	92.47
Pan et al. [45]	LDA + SVM	90.70	N/A	89.7	81.3
Li et al. [46]	IWLD + SVM	90.57	89.01	91.61	81.22
Our method	DHT + RoF	96.35	96.57	95.76	92.95

## Data Availability

The data presented in this study are openly available in https://github.com/jie-pan111/Biology (accessed on 14 May 2022).

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
