# Peer review of "A Novel Ensemble Learning-Based Computational Method to Predict Protein-Protein Interactions from Protein Primary Sequences"

_biology, 2022, doi:10.3390/biology11050775_

Round 1

Reviewer 1 Report

The manuscript by Jie Pan et al. describes a method of predicting protein-protein interactions using only sequence data. The method is based on DHT and RoF, was evaluated using three species, and compared to other machine learning methods and previously published methods. The authors demonstrate that their method is working well, and hence, I will recommend Biology to consider publishing this manuscript after major revisions as detailed below.

Major concerns:

  • Please provide access to scripts, software, ML models, and a README so the readers can reproduce the findings. Please use both a publicly available repository hosted on, for example, GitHub, and add a copy as supplementary material for the paper itself.
  • The authors reports on both a 1-D and a 2-D DHT in their methods section but only present the data for, presumably, 2-D in the results. Therefore, please expand the result section to include the data from the 1-D DHT.
  • The details regarding how the PSSMs were created need to be supplemented. Please provide details on what fasta database was used (and which version), how many rounds of PSI-BLAST were run, and other settings that might be important. PSI-BLAST is presumably the most expensive step in the algorithm, so please present the computational resources needed to execute this step.
  • While the method performs well on the pairs of PPIs in the article, the authors should discuss the practical limitations; at the very least, the authors should present false discovery rates if applied genome-wide for various organisms and the computational costs of producing PSSMs for each protein.

Minor concerns:

  • The language needs to be improved; some use of words and phrases is unconventional.

Author Response

Thank you for your precious comments and advice. Those comments are all valuable and very helpful for revising and improving our paper. We have revised the manuscript accordingly,and our point-by-point responses are presented above. Please see the attachment

Reviewer 2 Report

The work presented in this manuscript elaborates on a novel computational approach that aims to identify protein-protein interactions (PPIs) based on aminoacid sequence data. The process itself comprises of three steps; (1) a numerical transformation of the protein sequence data into a matrix using the Position Specific Scoring Matrix (PSSM), (2) a Discrete Hilbert (DH) transformation on the generated matrix in order to filter specific vectors as features, and (3) a Rotation Forest (RoF) i approach as the ensemble method to predict the PPIs. After a short introduction on the field of PPIs, and some of the main computational methods used to address this challenge with an emphasis on Machine-Learning based approaches, the manuscript continues with an overview of the main mathematical formulae behind the PSSM, the DH and the RF as part of the main Method presentation. The article concludes with an evaluation of the entire process, including an quantitative assessment of the model itself across the two datasets, a comparison of the process against classifiers other than RoF (i.e. SVM, KNN and RF), and an evaluation across four more datasets and external methods.

Overall the paper shows the potential for a novel approach towards the large-scale identification of PPIs. However, there are several points that should be clarified, in order to better understand both the process itself, as well as the evaluation strategy used.

Major comments

1. The main novelty of this work appears to be the unique combination of the PSSM representation of the sequence with the Hilbert transformation and the Rotation Forest ensemble method. However, the Methods section (and in particular Sections 2.2, 2.3 and 2.4) do not provide sufficient context into how each of these steps was applied into the actual data. In each case there is a detailed overview of the mathematical notation of each method (which appear to be the foundational knowledge in each case, and not developed specifically for this method), with very limited descriptions on how this is actually applied in practice to the data in consideration. Specifically:
1a: The PSSM matrix is produced through a multiple sequence alignment of sequences detected above a given score threshold using protein–protein BLAST. This, by definition, implies that each sequence of the input/training dataset is compared against a particular database, but there is no mention of such a database.
1b: Aside from a broad overview of the equations of the Discrete Hilbert transformation, the main connection to the previous step (i.e. the PSSM), is the statement that "For better capturing feature vectors from PSSM matrix, we applied the 2-D DHT that defined in the frequency domain to construct the local energy of PSSM" (line 142-143). However, this statement does not provide adequate information to clarify the assumptions used in representing a single-sequence matrix as a 2-D frequency representation. Moreover, the definition of a "local energy of PSSM" is not defined in this manuscript - and it also doesn't seem to be a term commonly used in literature. Finally, the statement "[...] the odd and even parts of PSSM features in frequency domain [...]" is rather unclear - do these refer to the order of appearance (i.e. the arbitrary order of the 20 aminoacids), or to the highly conserved order of aminoacids within a particular protein sequence?
1c: The application of RoF for the classification is absolutely valid, but it also implies the use of PCA in addition to the previous Hilbert transformation. The main risk in such an approach is the high potential for over-fitting, given that both matrix transformations lead to a bias towards the highly differentiated features. In that context, an assessment of (any) overfitting bias in this stage should be clarified. Finally, given that the RoF section (2.4) currently appears to not include any aspect that is unique to the work, the novelties introduced in this step (or the way it's applied to the data in question) should be highlighted more.

2. The description of the data used for the design of the process (section 2.1) is rather confusing. Given that CD-Hit is primarily a sequence clustering tool (based on a user-defined threshold which is usually the identity), the assumption is that the input dataset is filtered out for the unique, non-homologous, sequence pairs. However, there is no information on the number of pairs in the initial data, nor on the metrics and corresponding thresholds used to filter out the homology. Moreover, it's not clear why the application to the other PPI datasets (i.e. the Human and Rice PPIs datasets) establishes the generality of the method. If the whole process is applied in its entirety (i.e. from preprocessing to training and evaluation) in both cases, then any over-fitting issues will not be identified. If only the model generated by the 1st dataset was used, what are the unique aspects of the other datasets that are still captured by the generated model?

3. The evaluation section provides a very detailed overview of all metrics, and performs an admirable number of comparisons both across datasets and across methods. However, there are a few points that are unclear, specifically:
3a: The metrics reported under Section 3.2 are really impressive and clearly show that the model can effectively capture the PPIs. However, a 5-fold cross-validation, given the high-level of filtering introduced by the two pre-processing steps, is not sufficient to capture potential biases and / or model over-fitting issues. One approach could be to utilize the train/test/validation approach, using always a smaller subset of the training dataset completely unseen by the model process. In any case, a more formal ML reporting process (such as the DOME recommendations) might be useful here, in order to ensure that all aspects are adequately addressed.
3b: Comparing against different classifiers is absolutely useful - however, it should be mentioned the ensemble methods traditionally outperform single classifiers. This also supports the fact the the RF seems to occasionally outperform RoF. A comparison against other ensemble methods here would provide a much more accurate and effective comparison.

4. There is absolutely no statement of data and code availability. Given that the work presented here is primarily a computational approach, all code and data used (including the outputs generated) should be made available. Otherwise its virtually impossible to do an effective assessment of the method - nor will the tool itself be useful to the wider community. 

Minor points:
- line 117: where m denotes the length of the protein sequence. From the equation it's not clear whether m or b is the representative variable for the sequence length. 
- line 108: this sentence seems incomplete "After removing the sequence whose sequence identified bigger than 25%, and we construct 4262 negative samples."
- there are also a few typos and syntactic errors that should be addressed.

Author Response

Thank you for your letter and for the reviewers’ comments concerning our manuscript. Those comments are all valuable and very helpful for revising and improving our paper, as well as the important guiding significance to our researches. We have studied comments carefully and have made correction which we hope meet with approval. The main corrections in the paper and the responds to the reviewer’s comments are as flowing. Please see the attachment

Reviewer 3 Report

Overall the paper is well written with adequate methodologies.

  1. Overall, the authors talk about PPI`s in this paper and how to predict them. However they should define somewhere the definition of PPI in terms of how many proteins are involved. Does this paper predict a 2-protein or more proteins interaction? Can we predict a 3-4 protein interaction?
  2. In line 100, is there a upper limit?
  3. The conclusion is very short and can be improved as it just summarizes the results without further explanation or discussion.

Author Response

Thank you for your careful review. We really appreciate your efforts in reviewing our manuscript during this unprecedented and challenging time. We wish good health to you, your family, and community. Your careful review has helped to make our study clearer and more comprehensive. Please see the attachment.

Round 2

Reviewer 1 Report

The authors present a revised manuscript and made their code available. All of my concerns are addressed and I recommend Biology to consider this manuscript for publication.

Author Response

We sincerely hope that this revised manuscript has addressed all your comments and suggestions. We appreciated for reviewers’ warm work earnestly, and hope that the correction will meet with approval. Once again, thank you very much for your comments and suggestions.

Reviewer 2 Report

First of all, I would like to express my thanks to the authors for taking into consideration the comments raised - the manuscript is much clearer now in communicating the effort done. 

Although the vast majority of the concerns raised are adequately addressed, there are a few points that merit additional discussion (the same numbering of comments will be used, in order to maintain consistency across the revisions - any new comments are indicated with a "+"):

1a. It is now clear that the process of generating the PSSM is by running PSI-BLAST for each of the non-homologous sequences present in the PPI pairs, against the non-redundant SWISS-PROT database. Specifically, and to my understanding, for a given pair of proteins pi-pj from the PPI dataset, PSI-BLAST was run for each of the pi and pj independently to produce the two matrices Mi - Mj, which would be consequently used to represent the PPI. If that is indeed the case, it would be slightly better phrased in the text itself in order to effectively communicate the process. This also raises an additional issue;

1a+: PSI-BLAST has a number of parameters (beyond the number of iterations and the e-value cut-off). Specifically, the scoring matrix (BLOSUM* / PAM*) and the number of alignments used, have a major impact to the PSSM - with proteins that are highly conserved across all Domains to have higher scoring bias compared to others. To this end, it would be critical to list the explicit parameters used for running the tool itself.

1b. Although the explanation given in the response letter is perfectly adequate, the only change in the manuscript text is a replace of "[...] to construct the local energy of PSSM." with "[...] to extract features from the PSSM." It would be recommended that the text from the letter is also incorporated in the manuscript text as well - particularly the definition of the frequency domain with regards to the aminoacid positioning.

3. I would like to thank the authors for putting in the additional work of testing against a new classifier, as well as employing the train/test/validation process. However, regarding the latter, and although the response letter mentions that "We have tried to use the method (train/test/validation) you described for training predictions, but the predicted results did not improve.", the corresponding results are not present anywhere in the text. It should be highlighted here that the train/test/validation process aims to identify any potential over-/under-fitting issues with the models itself, and not to actually improve the prediction metrics. As such, if the overall metrics where poorer compared to the simple x-fold cross-validation process, might be an indication of over-fitting. To this end, it would be strongly recommended that these results are also listed in the manuscript as well - or at the very least as supplementary material.

4. The Matlab code used for the analysis is now available under GitHub (it would be advisable to also list a license, so that the wider community can reuse it). However, there are still three major issues with data/code availability:
4a+: The repository lacks any kind of structure, making it virtually impossible to understand the flow/order that the individual scripts should be executed.
4b+: There is no indication of the data used in each instance. Within the code itself there are references to .mat files (as well as hard-coded full paths), but the actual data is not accessible.
4c+: There is a significant pre-prossesing step, that includes the execution of PSI-BLAST and CD-Hit among others (always assuming that a command-line interface was used). The individual commands/scripts/parameters used for these steps are also necessary in order to effectively be able to reproduce the work done. If the CLI versions of the tools were not used, a clear documentation of the web-based/portal versions should be included.

Author Response

Authors’ Response to the Reviewers’ Comments

Paper title:

A novel ensemble learning-based computational method to predict protein-protein interactions from protein primary se-quence

Authors:

Jie Pan, Yan-Mei Sun, Chang-Qing Yu, Li-Ping Li, Zhu-Hong You, Shi-Wei Wang

We are grateful to the associate reviewers for putting in efforts to review the paper with the aim of improving the quality of our paper. We have addressed the concerns of the associate editor and reviewers in the newly revised manuscript. In particular, the following revisions have been made:

Reviewer 1

General comments to the authors:

First of all, I would like to express my thanks to the authors for taking into consideration the comments raised - the manuscript is much clearer now in communicating the effort done.

Although the vast majority of the concerns raised are adequately addressed, there are a few points that merit additional discussion (the same numbering of comments will be used, in order to maintain consistency across the revisions - any new comments are indicated with a "+"):

Response: Thanks the reviewer for the good summary and the comments regarding our article.

Major comments:

Comment 1:

1a. It is now clear that the process of generating the PSSM is by running PSI-BLAST for each of the non-homologous sequences present in the PPI pairs, against the non-redundant SWISS-PROT database. Specifically, and to my understanding, for a given pair of proteins pi-pj from the PPI dataset, PSI-BLAST was run for each of the pi and pj independently to produce the two matrices Mi - Mj, which would be consequently used to represent the PPI. If that is indeed the case, it would be slightly better phrased in the text itself in order to effectively communicate the process. This also raises an additional issue;

1a+: PSI-BLAST has a number of parameters (beyond the number of iterations and the e-value cut-off). Specifically, the scoring matrix (BLOSUM* / PAM*) and the number of alignments used, have a major impact to the PSSM - with proteins that are highly conserved across all Domains to have higher scoring bias compared to others. To this end, it would be critical to list the explicit parameters used for running the tool itself.

1b. Although the explanation given in the response letter is perfectly adequate, the only change in the manuscript text is a replace of "[...] to construct the local energy of PSSM." with "[...] to extract features from the PSSM." It would be recommended that the text from the letter is also incorporated in the manuscript text as well - particularly the definition of the frequency domain with regards to the amino acid positioning.

Response: We are grateful for the suggestion. to be more clearly and in accordance with the reviewer concerns, we have added a more detailed interpretation regarding these two questions.

1a+: We gratefully appreciate for your valuable suggestion. In this work, we chose three iterations and the e-value parameter was assigned to 0.001, and the PAM was selected as the scoring matrix. The other parameters were set to the default values. (Line: 151-152)

1b: Thanks for your comments. As suggested by reviewers, we add the definition of the frequency domain with regards to the amino acid positioning. (Line 171-174)

For better capturing feature vectors from PSSM matrix, we applied the 2-D DHT that defined in the frequency domain to extract features from the PSSM. The odd and even parts of PSSM features in frequency domain refer to the highly conserved order of amino acids within a particular protein sequence. (Line 171-174)

Comment 3:

I would like to thank the authors for putting in the additional work of testing against a new classifier, as well as employing the train/test/validation process. However, regarding the latter, and although the response letter mentions that "We have tried to use the method (train/test/validation) you described for training predictions, but the predicted results did not improve.", the corresponding results are not present anywhere in the text. It should be highlighted here that the train/test/validation process aims to identify any potential over-/under-fitting issues with the models itself, and not to actually improve the prediction metrics. As such, if the overall metrics where poorer compared to the simple x-fold cross-validation process, might be an indication of over-fitting. To this end, it would be strongly recommended that these results are also listed in the manuscript as well - or at the very least as supplementary material.

Response: Thank you for your suggestion. As suggested by reviewer, we listed these results based on the (train/test/validation) method on the supplementary material. (Line 293-295)

Table S1. The prediction results obtained by (train/test/validation) based method on the Yeast dataset.

Dataset

Acc (%)

Sens (%)

Spec (%)

PR (%)

MCC (%)

AUC

1

91.15

90.02

92.31

92.30

83.86

0.9533

2

91.64

90.13

93.14

92.88

84.67

0.9614

3

92.27

89.78

94.74

94.43

85.71

0.9602

4

92.31

90.55

94.15

94.18

85.80

0.9576

5

91.37

89.15

93.74

92.82

84.19

0.9598

Average

91.75±0.52

89.92±0.51

93.56±0.93

93.32±0.93

84.85±0.88

0.9585±0.0032

Table S2. The prediction results obtained by (train/test/validation) based method on the Human dataset.

Dataset

Acc (%)

Sens (%)

Spec (%)

PR (%)

MCC (%)

AUC

1

96.02

94.84

97.08

96.71

92.32

0.9832

2

95.47

95.11

95.81

95.59

91.34

0.9778

3

96.14

96.60

95.70

95.52

92.57

0.9852

4

96.75

95.19

98.07

97.67

93.66

0.9827

5

96.14

95.15

97.06

96.75

92.56

0.9881

Average

96.10±0.46

95.38±.0.70

96.75±0.99

96.45±0.90

92.49±0.83

0.9834±0.38

Table S3. The prediction results obtained by (train/test/validation) based method on the Oryza sativa dataset.

Dataset

Acc (%)

Sens (%)

Spec (%)

PR (%)

MCC (%)

AUC

1

93.85

94.64

93.12

92.84

88.64

0.9666

2

94.74

94.53

94.84

94.53

90.02

0.9677

3

94.84

84.80

94.59

94.80

90.22

0.9666

4

94.53

93.89

93.90

93.89

89.66

0.9659

5

94.06

95.32

94.97

95.32

88.81

0.9685

Average

94.41±0.43

94.55±0.78

94.28±0.77

94.28±0.99

89.43±0.77

0.9671±0.0010

Comment 4:

The Matlab code used for the analysis is now available under GitHub (it would be advisable to also list a license, so that the wider community can reuse it). However, there are still three major issues with data/code availability:

4a+: The repository lacks any kind of structure, making it virtually impossible to understand the flow/order that the individual scripts should be executed.

4b+: There is no indication of the data used in each instance. Within the code itself there are references to .mat files (as well as hard-coded full paths), but the actual data is not accessible.

4c+: There is a significant pre-prossesing step, that includes the execution of PSI-BLAST and CD-Hit among others (always assuming that a command-line interface was used). The individual commands/scripts/parameters used for these steps are also necessary in order to effectively be able to reproduce the work done. If the CLI versions of the tools were not used, a clear documentation of the web-based/portal versions should be included.

Response: We deeply appreciate the reviewer’s suggestion. According to the reviewer’s comment, we have reorganized the dataset and code and have re-uploaded it. (Line: 384-385)

Data Availability Statement: All code to reproduce our model can be obtained here https://github.com/jie-pan111/Biology

We sincerely hope that this revised manuscript has addressed all your comments and suggestions. We appreciated for reviewers’ warm work earnestly, and hope that the correction will meet with approval. Once again, thank you very much for your comments and suggestions.

Round 3

Reviewer 2 Report

First of all, many thanks to the authors for putting the effort and the time to address all points raised earlier.

All comments have been adequately addressed and I have no further concerns.

As a minor point, regarding the supplementary material, I would like to highlight the difference between the cross-validation and the train-test-validation approach. In principle, the latter includes a distinct, separate dataset (the test set) that is never taken into consideration during the model creation/training process. In this context, one could employ a cross-validation method for the hyperparameter optimization and then, using the optimal model that was generated through this process, create an unbiased assessment of the model by testing against the test set. Based on that, it would be recommended to include for each dataset in the supplementary table both the metrics of the training process and the metrics of the corresponding validation process. As it stands, it's not clear what these metrics actually capture.

Author Response

Thank you very much for giving us an opportunity to revise our manuscript.
